

# Lateral presentation of faces alters overall viewing strategy

Christopher J. Luke and Petra M.J. Pollux

School of Psychology, University of Lincoln, Lincoln, United Kingdom

## ABSTRACT

Eye tracking has been used during face categorisation and identification tasks to identify perceptually salient facial features and infer underlying cognitive processes. However, viewing patterns are influenced by a variety of gaze biases, drawing fixations to the centre of a screen and horizontally to the left side of face images (left-gaze bias). In order to investigate potential interactions between gaze biases uniquely associated with facial expression processing, and those associated with screen location, face stimuli were presented in three possible screen positions to the left, right and centre. Comparisons of fixations between screen locations highlight a significant impact of the screen centre bias, pulling fixations towards the centre of the screen and modifying gaze biases generally observed during facial categorisation tasks. A left horizontal bias for fixations was found to be independent of screen position but interacting with screen centre bias, drawing fixations to the left hemi-face rather than just to the left of the screen. Implications for eye tracking studies utilising centrally presented faces are discussed.

## INTRODUCTION

Eye-movements provide a way of measuring attention and can highlight perceptually salient facial features for facial identity and expression recognition (*Jack et al.*, *2009*). Viewing patterns toward faces have been well documented. First fixations exhibit a centre-of-face bias which has been interpreted as object selection (*Foulsham & Kingstone*, *2013*; *Levy, Foulsham & Kingstone*, *2013*) and the first stage of expression recognition, allowing rapid early analysis of expression (*Calvo, Nummenmaa & Avero*, *2008*; *Eisenbarth & Alpers*, *2011*; *Feldmann-Wüstefeld, Schmidt-Daffy & Schubö*, *2011*; *Guo*, *2012*; *Hills, Cooper & Pake*, *2013*; *Pollux, Hall & Guo*, *2014*; *Samson et al.*, *2014*). Visual search tasks have been used to demonstrate that the initial fixation landing position on faces is decided during pre attentive processing and is used to overtly orient attention and allocate attentional resources when processing the face (*Calvo, Nummenmaa & Avero*, *2008*). The initial central fixation is followed by a strong focus on the eyes and mouth, which are considered as the most diagnostic facial features for categorisation of different facial expressions (*Calvo, Nummenmaa & Avero*, *2008*; *Eisenbarth & Alpers*, *2011*; *Kohler et al.*, *2004*; *Levy, Foulsham & Kingstone*, *2013*; *Maurer, Grand & Mondloch*, *2002*; *Messinger et al.*, *2012*; *Rigato & Farroni*, *2013*; *Smyth et al.*, *2005*; *Vassallo, Cooper & Douglas*, *2009*; *Wang et al.*, *2011*; *Xiao et al.*, *2013*) or for identity recognition (*Sæther et al.*, *2009*; *Van Belle et al.*, *2010*).

Corresponding author
Christopher J. Luke,
cluke@lincoln.ac.uk

Preferential feature selection varies between emotions (*Eisenbarth & Alpers*, *2011*; *Pollux, Hall & Guo*, *2014*) and culture (*Jack et al.*, *2009*) but predominantly focuses on the eye region, which is selected early and frequently for fixations (*Eisenbarth & Alpers*, *2011*; *Levy, Foulsham & Kingstone*, *2013*; *Samson et al.*, *2014*). Fixations towards the eyes are independent of their position in the face, as demonstrated in a study using monsters with non-typical eye locations (*Levy, Foulsham & Kingstone*, *2013*). Eyes located in the centre of a face or peripherally located on limbs were fixated quickly and frequently, showing that the eyes themselves are the focus of attention and not their relative position on the face. Early selection of the eyes is not only attributed to emotion categorisation, and is seen as extraction of socially relevant information from the face (*Gobel, Kim & Richardson*, *2015*; *Levy, Foulsham & Kingstone*, *2013*).

The initial centre-of-face bias in gaze behaviour is commonly observed in studies where face stimuli are presented in the centre of the screen (*Guo*, *2012*; *Levy, Foulsham & Kingstone*, *2013*; *Pollux, Hall & Guo*, *2014*; *Rigato & Farroni*, *2013*; *Samson et al.*, *2014*). However, evidence from natural scenes shows that when presented with landscapes on a screen, observers generally make the first fixation to the centre of the display (*Bindemann*, *2010*). This central tendency is not limited to first fixations: eye movement patterns tend to exhibit a gravitational pull towards the screen centre throughout the viewing period (*Tatler*, *2007*). Central tendency for fixations is also resistant to the distribution of features in natural scenes (*Tatler*, *2007*) and to manipulations of the central fixation marker, for example by displaying it peripherally on a screen in any number of locations (*Bindemann*, *2010*). Similarly, moving the position of the entire screen to the left or right of an observer's natural viewing position does not eliminate a screen centre bias (*Vitu et al.*, *2004*). The potential role of the central screen bias on gaze patterns during face viewing for emotion expression categorization has not been investigated systematically. Given the robust nature of this bias, it is not clear whether the centre-of-face bias, previously associated with rapid extraction of diagnostic facial features for emotion recognition (*Calvo, Nummenmaa & Avero*, *2008*; *Eisenbarth & Alpers*, *2011*; *Feldmann-Wüstefeld, Schmidt-Daffy & Schubö*, *2011*; *Guo*, *2012*; *Hills, Cooper & Pake*, *2013*; *Levy, Foulsham & Kingstone*, *2013*; *Pollux, Hall & Guo*, *2014*; *Samson et al.*, *2014*), could be attributed to the central position of face images on the screen in previous studies (*Guo*, *2012*; *Levy, Foulsham & Kingstone*, *2013*; *Pollux, Hall & Guo*, *2014*; *Rigato & Farroni*, *2013*; *Samson et al.*, *2014*).

A second gaze bias associated with face viewing is the tendency to preferentially view the left hemi-face, from an observers perspective (*Guo*, *2012*) or faces presented in the left visual field (*Prete et al.*, *2015*), which has been suggested to specifically benefit categorisations of facial expression. Evidence of facial muscles portraying emotions more intensely in the left hemi-face (*Indersmitten & Gur*, *2003*) suggests that more diagnostic information is available on the left, which *Indersmitten & Gur* (*2003*) propose is due to a right hemispheric dominance for emotion processing. The argument is supported by evidence showing that the left side of the face is less subject to cultural influences, presenting a more universally recognised display of emotional expressions (*Mandal & Ambady*, *2004*). However, evidence from natural scenes challenges a face specific left gaze bias, demonstrating a general horizontal bias to the left visual field (*Foulsham et al.*, *2013*; *Ossandón, Onat & König*, *2014*).
Similarly, when saccading toward objects, observers typically undershoot their target slightly to the left (*Foulsham & Kingstone*, *2013*). Methodological factors have also been shown to influence left gaze bias, which is entirely negated for face viewing during a gender judgement task when faces are presented on either side of an initial fixation point (*Samson et al.*, *2014*). In these conditions, participants preferentially view the hemi-face closest to the fixation point, suggesting that left gaze bias may be an artefact of central stimulus presentation. Furthermore, during a free viewing task where time constraints were not introduced, participants did not demonstrate a bias to either side of the face, an effect the authors propose to be related to long exploration periods balancing out an initial left processing bias (*Eisenbarth & Alpers*, *2011*).

In order to accurately assess viewing patterns attributed to facial expression categorisation we aim to dissociate generic or methodological gaze biases associated with the use of a screen from face specific biases, by directly comparing viewing patterns between centrally and laterally presented stimuli. Specific biases to be investigated include the central gravitational bias for fixations (*Bindemann*, *2010*; *Foulsham et al.*, *2013*; *Ossandón, Onat & König*, *2014*; *Tatler*, *2007*), which would result in a higher number of fixations to the centre of the face only in centrally presented images and to the hemi-face proximal to the screen centre in laterally presented images. Three emotions will be shown, happy, sad and fear, as the nose regions for these expressions are generally not considered to be crucially diagnostic for correct categorization (*Calvo, Nummenmaa & Avero*, *2008*; *Eisenbarth & Alpers*, *2011*; *Ekman & Friesen*, *1978*; *Kohler et al.*, *2004*; *Levy, Foulsham & Kingstone*, *2013*; *Maurer, Grand & Mondloch*, *2002*; *Messinger et al.*, *2012*; *Rigato & Farroni*, *2013*; *Smyth et al.*, *2005*; *Vassallo, Cooper & Douglas*, *2009*; *Wang et al.*, *2011*; *Xiao et al.*, *2013*). Any central fixation biases are therefore more likely attributable to screen biases. The second bias under investigation is the left gaze bias (*Bindemann*, *2010*; *Foulsham et al.*, *2013*; *Guo*, *2012*). Specifically, the impact of lateral presentation and the absence of imposed time constraint is expected to diminish or eliminate a bias to the left side of the face (*Eisenbarth & Alpers*, *2011*) but not to the left side of the screen (*Bindemann*, *2010*; *Foulsham et al.*, *2013*).

## METHODS

### Participants

To control for a potential gender bias (*Hall*, *1978*; *Vassallo, Cooper & Douglas*, *2009*) only female participants were included who typically perform better at emotion recognition tasks (*Wang*, *2013*); twenty one undergraduate students from the University of Lincoln took part in the experiment (21 female, mean age = $19.19 \pm 1.03$). All participants had normal or corrected to normal visual acuity at the time of testing, received no instructions on eye movements and completed an informed consent form prior to taking part in a single session lasting approximately 25 min. The experiments were granted ethical approval from the School of psychology research ethics committee at the University of Lincoln.

## Apparatus

A Tobii T60XL widescreen eye tracker served as eye tracker and monitor displaying at 1,280 × 1,024 pixels at a refresh rate of 60 Hz, stimuli were presented at a size of 900 × 550 pixels subtending a visual angle of 23.110 and 11.674° respectively. Matlab with Psychtoolbox and the Tobii Matlab Toolbox were used for visual stimulus control and to run the eye tracker. The gaze precision of the eye tracker is reported at 0.5 visual degrees with binocular sampling at a distance of 65 cm. Fixations were computed using a dispersion algorithm (*Salvucci & Goldberg*, *2000*). Behavioural responses were collected using a Cedrus RB-540 response pad.

## Stimuli

Stimuli were generated using the Karolinska Directed Emotional Faces database (*Lundqvist, Flykt & Öhman*, *1998*). Two male (AM10, AM23) and two female models (AF01, AF09) were chosen displaying prototypical expressions of happy, sad and fear. Images were converted to grey-scale and balanced for contrast and brightness; extraneous features such as hair, ears and neck were removed by placing an oval frame around the face. In order to manipulate task demand and avoid ceiling performance, emotions were morphed between neutral and emotional using the Morpheus Photomorphing Suite, creating ten intensity stages, labelled from 10% to 100%. Based on previous findings of improvements in expression recognition only at low intensities and a ceiling to performance increases beyond 50–60% (*Gao & Maurer*, *2010*; *Pollux, Hall & Guo*, *2014*; *Pollux*, *2016*) intensities of 70, 80 and 90% were removed whereas 100% was included to control for task comprehension, leaving a total of 84 images used in the study.

## Procedure

Stimuli were presented three times, once per location on screen; possible screen locations were to the left, right or centre. Screen locations were centered on quartile pixel calculations of the $x$ axes of the screen, for example left presented faces centered on pixel 320 ($\frac{1280}{4}$). Participants were seated 65 cm away from the monitor and used their preferred hand to make responses; calibration required participants to focus on the centre of a shrinking dot randomly presented in sequence using a 5 point calibration array. The main task required participants to quickly and accurately categorise displayed facial expressions according to three possible responses, happy, sad or fear, though no time limit was imposed. Each trial's screen position was randomly chosen and stimuli were presented in a random order based on selected screen position; each stimulus appeared once per location. After an instruction screen, each trial commenced with a fixation cross presented centrally for 500 ms, followed by a facial stimulus at one of the three locations. The stimulus remained on screen until a participant pressed any response key to indicate that they recognized the emotion. After this key press, a choice selection screen detailing the possible responses and the corresponding keys. This procedure was chosen due to the number of possible responses, to eliminate button selection time from the viewing period.

## RESULTS

Initial analysis included comparisons for stimuli gender and participant ocular dominance, measured using the Dolman method (*Cheng et al.*, *2004*). However, no effect was found on proportion correct responses or eye movements. Therefore, these factors were not included in further analysis.

Reaction times (RT's) were analysed by entering average RT's into a $3 \times 3 \times 7$ Repeated Measures ANOVA (Emotion × Screen position × Intensity). Bonferroni corrected pairwise comparisons were used to compare main effects and Greenhouse Geisser adjustment was used where appropriate. Results showed no significant differences in RT's between each of the possible screen positions ($F(2,40) = 0.359$, $p = 0.701$, $\eta p^2 = 0.018$), however the main effect of emotion ($F(2,40) = 6.754$, $p = 0.003$, $\eta p^2 = 0.252$) was significant due to fear expressions being responded to faster (mean 1,454ms) compared to sad expressions (mean 1,660 ms, $p = 0.010$). Intensity was also significant ($F(6,120) = 9.582$, $p < 0.001$, $\eta p^2 = 0.324$) as lower intensities were responded to slower than higher intensities. 10% intensity (mean 1,866 ms) was responded to significantly slower than 100% (mean 1,241 ms, $p = 0.045$), 20% (mean 1,780 ms) was responded to significantly slower than 60% (mean = 1,375 ms, $p = 0.042$) or 100% ($p = 0.041$) and 40% intensity (mean 1,575 ms) was responded to significantly slower than 50% (mean = 1,399 ms, $p = 0.043$), 60% ($p = 0.037$) and 100% ($p = 0.006$).

Accuracy was analysed by entering percentage correct responses for each screen position into a $3 \times 3 \times 7$ Repeated Measures ANOVA (Emotion × Screen position × Intensity). Bonferroni corrected pairwise comparisons were used to compare main effects and Greenhouse Geisser adjustment was used where appropriate. Results showed no significant differences in accuracy between each of the three Screen positions ($F(2,40) = 0.596$, $p = 0.556$, $\eta p^2 = 0.029$); average correct response across all three screen positions was $74 \pm 1\%$. Emotion did have a significant effect on accuracy ($F(2,40) = 40.191$, $p < 0.001$, $\eta p^2 = 0.668$) which was due to sad expressions being correctly categorised (mean = 89%) more than happy (mean = 71%) or fear (mean = 63%, $p$'s $< 0.001$). Intensity was also significant ($F(6,120) = 27.615$, $p < 0.001$, $\eta p^2 = 0.580$), improvements in categorisation performance were seen from 10% intensity (mean 50% correct) to 20% (mean 61%), and 30% (mean 71%) to 40% (mean 79%). At high intensities there were no significant differences of categorisation performance, though the trend to increase performance continued (10% < 20%/30% < 40%/50%/60%/100%, $p$'s $< 0.011$). Finally, emotion and intensity interacted ($F(12,240) = 11.515$, $p < 0.001$, $\eta p^2 = 0.365$). Compared to sad (range = 7%, $p$'s $> 0.913$), for which accuracy did not change significantly from low intensity to high intensity, fear (range = 60%, 10% < 20%/30% < 40%/50%/60%/100%, $p$'s $< 0.004$) and happy (range = 48%, 10% < 40%/50%/60%/100%, $p$'s $< 0.022$) had larger improvements from low intensity to high intensity.

Face viewing was measured by defining three regions of interest (ROI); the eyes, the nose and the mouth. The eyes ROI included the brows, upper and lower lids and a surrounding area of approximately 2 visual degrees. The nose ROI included the bridge, nasal root and a surrounding area up to 2 visual degrees where this did not impact on other ROI's. Finally

the mouth ROI included the lips, mentolabial sulcus and philtrum and a surrounding area of 2 visual degrees. Each ROI was designed to encompass the face accurately for any expression at all intensities so that gaze biases introduced by the screen or stimulus position would not impact on analyses between expressions. Fixations that were not within the boundaries of the displayed image ($900 \times 550$ pixels) were removed from analysis, however, fixations within the image but not within any ROI (eyes, nose or mouth) were included in total fixation calculations. Therefore, the total number of fixations for each stimulus were comprised of fixations to the three ROI's being analysed, as well as fixations to any other area of the face.

## Central bias

Due to the free viewing method each stimulus received a variable total number of fixations, therefore these were converted to percentage of total fixations for comparisons between participants and across stimuli, percentage fixations were then averaged across the four models. To analyse the effect of screen position (3), emotion (3) and intensity level (7) on the linear combination of 'Percentage Fixations' (on mouth, eyes and nose), these percentages were entered in a 3 (Screen position) $\times$ 3 (Emotion) $\times$ 7 (Intensity) Repeated Measures MANOVA.

Multivariate analysis revealed that screen position was significant [Wilk's $\lambda = 0.107$, $F(6,76) = 26.575$, $p < 0.001$, $\eta p^2 = 0.677$] and univariate analysis showed that the percentage of fixations to eyes ($F(2,40) = 5.64$, $p = 0.007$, $\eta p^2 = 0.220$), nose ($F(2,40) = 100.98$, $p < 0.001$, $\eta p^2 = 0.835$) and mouth ($F(2,40) = 35.29$, $p < 0.001$, $\eta p^2 = 0.638$) all varied significantly dependent on screen position.

Bonferroni corrected pairwise comparisons were used to analyse main and interaction effects. The effect of Screen position was compared separately for each ROI (see Fig. 1). Percentage fixations to the eyes varied significantly between faces presented in the centre and to the right ($p$'s < or equal to 0.003) of the screen. The nose was fixated less when faces were presented to the right and left compared to those presented centrally in the screen ($p$'s < 0.001). Percentage fixations to the mouth were significantly different for all three screen positions (for all comparisons, $p$'s < 0.05). A multivariate interaction between Screen position and Intensity [Wilk's $\lambda = 0.713$, $F(36,703.925) = 2.373$, $p < 0.001$, $\eta p^2 = 0.107$], which was accounted for by the univariate interaction between Screen position and Intensity for the nose region ($F(12,240) = 3.870$, $p < 0.001$, $\eta p^2 = 0.162$) and the mouth region ($F(12,240) = 2.411$, $p = 0.006$, $\eta p^2 = 0.108$), suggest that the effect of screen position for nose and mouth in Fig. 1 was not exactly the same at all intensity levels. Most pairwise comparisons confirmed the effects illustrated in Fig. 1: there were more fixations to the nose in centrally presented faces at all intensities compared to left ($p$'s < 0.001) or right ($p$'s < 0.001) presented faces and centrally presented faces had fewer fixations to the mouth at all intensities compared to left ($p$'s < 0.001) or right presentations ($p$'s < 0.038). However, when left and right screen positions are directly compared, the nose was fixated more in left compared to right presented faces at 100% intensity ($p = 0.049$) and the mouth was viewed more in left compared to right presented faces at intensity levels 30% ($p = 0.001$) and 100% intensities ($p = 0.020$).

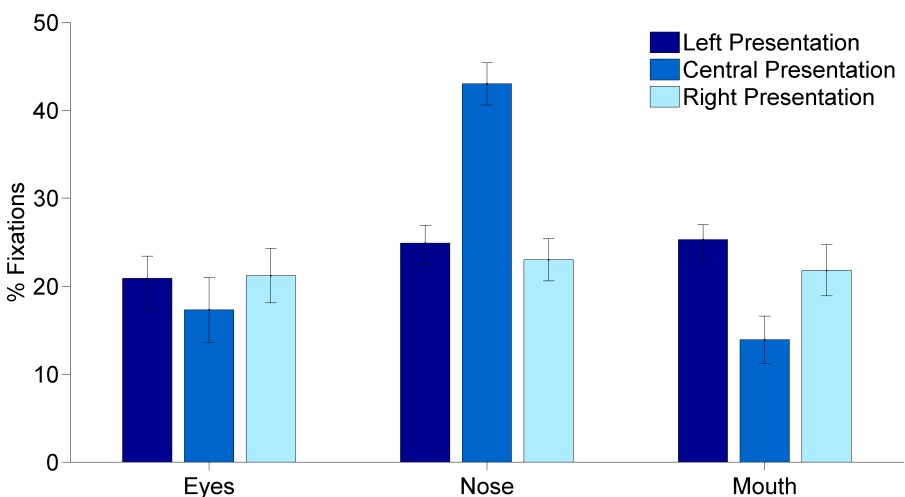

**Figure 1** Percentage fixations to predefined regions of interest, eyes, nose and mouth dependent on face presentation position on screen, left centre or right.

A significant multivariate effect of emotion [Wilk's $\lambda = 0.527$, $F(6,18) = 4.775$, $p < 0.001$, $\eta p^2 = 0.274$] was accounted for by a significant univariate effect of emotion for percentage fixations to the mouth only ($F(2,76) = 15.96$, $p < 0.001$, $\eta p^2 = 0.444$). Pairwise comparisons showed this was due to significant differences between all three emotions, with happy receiving the highest percentage of fixations to the mouth (mean = 22.253), fear receiving fewer (mean = 20.627) and sad receiving the lowest percentage (mean = 18.1191, all $p$'s < 0.020). Furthermore the multivariate effect of Intensity [Wilk's $\lambda = 0.691$, $F(18,334.240) = 2.588$, $p < 0.001$, $\eta p^2 = 0.116$] was accounted for by the univariate effect of Intensity on percentage fixation toward the eyes ($F(6,120) = 4.539$, $p < 0.001$, $\eta p^2 = 0.185$). Pairwise comparisons showed that fixations towards the eyes were higher at intensity 10% (mean = 21.80) compared to 30% (mean = 19.00, p = 0.016) or 60% (mean = 18.10, $p = 0.018$).

A significant multivariate interaction effect between Screen position and Emotion [Wilk's $\lambda = 0.648$, $F(12, 206.660) = 3.066$, $p = 0.001$, $\eta p^2 = 0.135$] was found, which was accounted for by a significant interaction between Screen Position and Emotion for the eye-region only ($F(4,80) = 5.264$, $p = 0.001$, $\eta p^2 = 0.208$). Pairwise comparison found fewer fixations to the eyes of fear expressions that were centrally presented compared to right presentations ($p = 0.043$). Similarly, sad expressions received fewer fixations to the eyes when centrally presented, compared to right ($p = 0.001$) presentations. Comparing emotions within each screen position pairwise comparisons found that fear expressions had more fixations to the eyes (mean = 23%) than happy (mean = 20%, $p = 0.004$) or sad expressions (mean = 20%) only when faces were presented to the right of the screen, all other comparisons were not significant.

## Left horizontal bias

To investigate left or right face or screen biases, percentage fixations within the face were calculated as percentages of those to the left hemi-face and those to the right hemi-face in

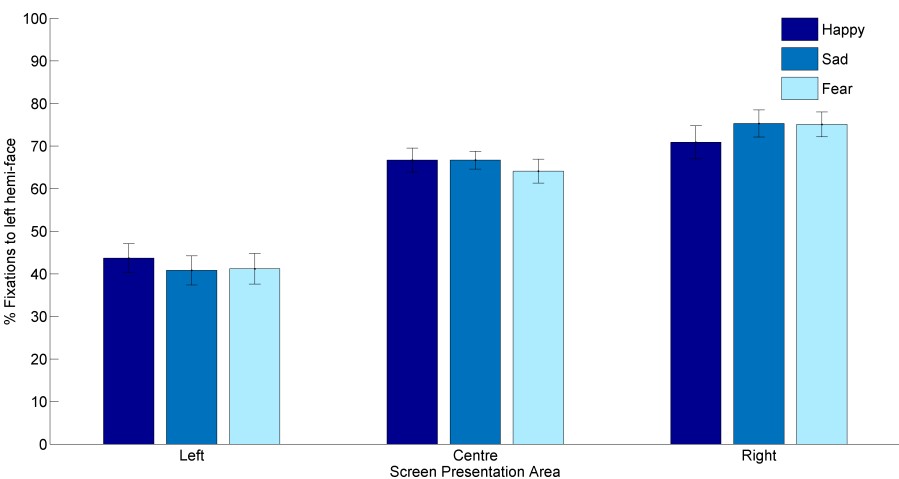

**Figure 2** Percentage fixations to the left hemi-face in each of the three screen presentation areas (left, centre and right) for each emotion (happy, sad and fear).

each of the three screen positions, with left and right hemi-face fixations equalling the total number of fixations made during stimulus presentation. Percentage of fixations to the left and to the right hemi-faces were averaged across the four models shown for each emotion and intensity, then these average percentages were entered into two 3 (Screen position) × 3 (Emotion) × 7 (Intensity) Repeated Measures ANOVAs, for separate analyses of percentage fixations to the left side and right side of the face.

A significant effect of Screen position was found for fixations to the left hemi-face ($F(2,40) = 100.067$, $p < 0.001$, $\eta p^2 = 0.833$) and to the right hemi-face ($F(2,40) = 135.155$, $p < 0.001$, $\eta p^2 = 0.871$). Figure 2 shows that the number of fixations to the left hemi-face increased as the image screen position changed to the right of the screen and therefore conversely that the number of fixations to the right hemi-face reduced. Pairwise comparisons showed significant differences between all three screen positions, for fixations to both the left ($p$'s $< 0.001$) and right hemi-face ($p$'s $< 0.001$).

Significant interaction effects were further found between Screen position and Emotion [left hemi-face: $F(4,80) = 4.337$, $p = 0.003$, $\eta p^2 = 0.178$. right hemi-face: $F(4,80) = 4.684$, $p = 0.002$, $\eta p^2 = 0.190$]. Fixations to the left hemi-face were significantly lower for fear expressions presented on the left compared to the right or centre ($p$'s $< 0.001$). For both happy and sad expressions, fixations to the left hemi-face were lower on faces presented to the left compared to the centre ($p$'s $< 0.001$) and higher on faces presented to the right compared to centre ($p$'s $<$ or equal 0.003, see Fig. 2). Conversely fixations to the right hemi-face were lowest for all emotions when faces were presented to the right compared to centrally or to the left, and highest for faces presented to the left compared to the centre or right (all $p$'s $< 0.001$). Within each screen position, fixations to the left hemi-face varied significantly only between happy (mean = 43%) and fearful expressions (mean = 41%, $p = 0.046$) in left presentations and happy (mean = 70%) and sad (mean = 75%, $p = 0.008$) and happy and fearful (mean = 75%, $p = 0.021$) expressions in right presentations. Fixations to the right hemi-face when stimuli were presented in the centre of the screen

were higher for sad expressions (mean = 23%) compared to happy (mean = 21%, $p = 0.018$) or fearful expressions (mean = 20%, $p < 0.001$) but when stimuli were presented to the left fearful expressions received more fixations to the right hemi-face (mean = 46%) than happy expressions (mean = 43%, $p = 0.046$). Finally, right hemi-face fixations were lower for sad expressions when stimuli were presented on the right of the screen (mean = 11%) compared to happy (mean = 13%, $p = 0.008$) or fearful expressions (mean = 13 $p = 0.006$). Emotion and Intensity [left hemi-face: $F(12,240) = 2.988$, $p = 0.017$, $\eta p^2 = 0.130$, right hemi-face: $F(12,240) = 2.815$, $p = 0.001$, $\eta p^2 = 0.123$] revealed that at 40% intensity, sad expressions had more fixations to the right hemi-face than fear ($p = 0.006$) and fewer fixations to the left hemi-face compared to happy ($p = 0.031$); happy expressions had fewer right hemi-face fixations compared to fear at 10% intensity ($p = 0.003$).

Finally, Screen position, Emotion and Intensity was significant for fixations to the left hemi-face only ($F(24,480) = 3.762$, $p = 0.003$, $\eta p^2 = 0.158$). Pairwise comparison showed that when presented centrally, all emotions at all intensities had more left hemi-face fixations than when presented on the left ($p$'s < or equal 0.048). Faces presented on the right of the screen also had more left hemi-face fixations than those presented to the left ($p$'s < or equal 0.021) except fear at 30% which did not vary significantly between right and left presentations. Right presented faces typically had more left hemi-face fixations than centrally presented faces, this was significant for fear expressions at 40% intensity ($p = 0.002$), happy expressions at 20% ($p = 0.002$), 30% ($p = 0.014$), 50% ($p = 0.006$) and 60% intensity ($p = 0.029$) and finally, for sad expressions at 20% ($p = 0.003$), 30% ($p = 0.002$) and 50% ($p < 0.001$) intensity. Next, when faces were presented in the centre of the screen happy expressions received more left hemi-face fixations than fearful expressions at 10% and 30% intensities ($p$'s < 0.036) and happy expressions also had more left hemi-face fixations than sad faces at 20% and 30% intensities ($p$'s < 0.048). Sad expressions also received more left hemi-face fixations than fearful expressions at 20% and 40% intensities ($p$'s < 0.028). When faces were presented to the left of the screen happy expressions had more left hemi-face fixations than sad at 30% and 50% intensities ($p$'s < 0.049) and fear at 30% ($p = 0.018$). Sad expressions also had more left face fixations than fear at 20% and 50% ($p$'s < 0.045) but fewer fixations than fear at 60% intensity ($p = 0.049$). Lastly, when faces were presented to the right of the screen happy expressions had fewer left hemi-face fixations than sad at 10%, 20%, 50% and 100% ($p$'s < 0.030) but more left hemi-face fixations than sad at 40% ($p = 0.033$). Happy expressions also had fewer left face fixations than fear at 20% and 30% ($p$'s < 0.009) but more left face fixations than fear at 60% ($p = 0.032$), sad expressions only had more left face fixations than fear at 60% intensity ($p = 0.018$).

## DISCUSSION

The present study was designed to differentiate general screen biases in viewing from those associated specifically to faces during categorisation tasks, in particular a tendency for fixations to focus around the centre of the face (*Guo, 2012*; *Levy, Foulsham & Kingstone, 2013*; *Pollux, Hall & Guo, 2014*; *Rigato & Farroni, 2013*; *Samson et al., 2014*) and for fixations to land on the left hemi-face (*Guo, 2012*). Stimulus screen position had a

significant impact on participants fixation patterns toward faces, specifically, laterally presenting faces on either side of a screen resulted in a large reduction in overall fixations towards the centre of the face when compared to centrally presented faces. Furthermore, the gravitational effect of screen centre on fixations (*Tatler, 2007*) was demonstrated by an increase in fixations to the hemi-face closest to screen centre even in laterally presented stimuli. This suggests that the centre of screen bias observed in studies using natural scenes (*Bindemann, 2010*; *Tatler, 2007*; *Vitu et al., 2004*) extends to face viewing and that the previously observed preference for face centre throughout viewing (*Guo, 2012*; *Levy, Foulsham & Kingstone, 2013*; *Pollux, Hall & Guo, 2014*; *Rigato & Farroni, 2013*; *Samson et al., 2014*) could be attributed to a general viewing bias introduced by the screen. In contrast, the left-gaze bias for faces (*Guo, 2012*) was not solely attributable to general screen biases as left-gaze persisted regardless of stimulus screen position, though this interacted with the centre of screen bias resulting in even fixations to each hemi-face in stimuli presented to the left of the screen. This finding is in contrast with previous evidence showing elimination of the left-gaze bias when faces are displayed laterally (*Samson et al., 2014*) and extended viewing periods are allowed (*Eisenbarth & Alpers, 2011*) but is compatible with a tendency to preferentially select the left side of objects (*Foulsham & Kingstone, 2013*) or faces presented to the left (*Prete et al., 2015*).

Displayed emotions were chosen specifically to contain little or no informative facial characteristics in the nose region, with fear displaying primarily in the eyes and happiness and sadness displaying primarily in the eyes and mouth (*Calvo, Nummenmaa & Avero, 2008*; *Eisenbarth & Alpers, 2011*; *Ekman & Friesen, 1978*; *Kohler et al., 2004*; *Levy, Foulsham & Kingstone, 2013*; *Maurer, Grand & Mondloch, 2002*; *Messinger et al., 2012*; *Rigato & Farroni, 2013*; *Smyth et al., 2005*; *Vassallo, Cooper & Douglas, 2009*; *Wang et al., 2011*; *Xiao et al., 2013*). The screen centre bias for landscapes and objects is suggested to arise from perceiving the screen itself as an object, which are also typically fixated at the centre (*Bindemann, 2010*; *Foulsham & Kingstone, 2013*). Our finding of a strong centre of screen bias, shown by fixations to the nose in centrally presented stimuli and fixations to the hemi-face closest to screen centre, supports the screen being perceived as an object (*Foulsham & Kingstone, 2013*) where fixations are typically drawn to the centre of the object being viewed. The central bias was reduced considerably when faces were laterally presented, reflected in a more balanced percentage of fixations across the three defined regions of interest. However, fixations toward the nose were not eliminated entirely, suggesting that details in the nose region were informative for categorization responses. Alternatively, fixations in this region may have been associated predominantly with early stages of face-viewing and could have been a reflection of a centre-of-face bias, aiding rapid early expression analysis (*Calvo, Nummenmaa & Avero, 2008*; *Eisenbarth & Alpers, 2011*; *Feldmann-Wüstefeld, Schmidt-Daffy & Schubö, 2011*; *Guo, 2012*; *Hills, Cooper & Pake, 2013*; *Pollux, Hall & Guo, 2014*; *Samson et al., 2014*). Future studies will be required to explore whether different viewing biases exert stronger influences at early and later stages of face viewing for expression categorization.

Our data shows that a screen centre bias, reflected in preferential attending of the hemi-face closest to screen centre, co-occurs with left hemi-face bias. Faces presented to

the left of the screen had a similar percentage of fixations to the left hemi-face and right hemi-face, whereas faces presented to the right of the screen received around six times more fixations to the left hemi-face compared to the right hemi-face. Due to the influence of a screen centre gravitational effect (*Tatler*, *2007*), fixations to faces presented on the left would be expected to fall primarily on the right hemi-face as previously observed (*Samson et al.*, *2014*). However, participants viewed both hemi-faces equally during left presentation, showing the influence of the left hemi-face bias drawing fixations to the left hemi-face whilst the screen centre bias concurrently draws fixations to the right hemi-face. In contrast, the two biases significantly increase fixations to the left hemi-face in right screen presentations. *Samson et al.* (*2014*) utilised restricted viewing time to control the total number of saccades participants could make, whereas here we utilised a free viewing task allowing unlimited visual exploration of the face. In both instances, a screen centre bias was observed, drawing fixations to the hemi-face closest to screen centre. Unlike *Samson et al.* (*2014*) we also observed a left hemi-face bias, drawing fixations to the left side of the face. Differences between our findings and those of *Samson et al.* (*2014*) cannot be due to viewing time, as previous studies have demonstrated that free viewing can eliminate a bias to the left hemi-face (*Eisenbarth & Alpers*, *2011*; *Voyer, Voyer & Tramonte*, *2012*). Therefore, the appearance of left gaze bias in our task remains only as a characteristic of emotion categorisation, as *Eisenbarth & Alpers* (*2011*) utilised valence and arousal rating scales rather than emotion categories and *Samson et al.* (*2014*) utilised a gender judgement task while *Levy et al.* (*1983*) asked participants to judge happiness in pairs of chimeric faces.

In addition to centre-of-screen and left hemi-face gaze biases, the results of the present study seem to suggest that a small bias towards the left compared to the right side of screen may have influenced gaze patterns, although this effect was small and only observed for the nose and mouth and was restricted to only a few intensity levels. However, this trend is consistent with the horizontal left bias previously reported in free viewing of natural scenes (*Foulsham et al.*, *2013*; *Foulsham & Kingstone*, *2013*; *Ossandón, Onat & König*, *2014*) and faces (*Prete et al.*, *2015*) and may warrant further exploration in future studies. If, as suggested in the present study, this bias is relatively small compared to the centre-of-screen and left hemi-face bias, then it may require experiments with a larger number of trials per intensity level to reveal the nature of this bias in facial expression recognition experiments.

In summary, a bias to the left hemi-face for fixations was dissociable from a general left horizontal bias (*Foulsham et al.*, *2013*; *Prete et al.*, *2015*) specifically as a characteristic of emotion categorisation tasks. The left hemi-face bias co-occurred with a screen centre bias (*Bindemann*, *2010*; *Ossandón, Onat & König*, *2014*; *Tatler*, *2007*), drawing fixations gravitationally towards the centre of the display screen whilst simultaneously drawing fixations to the left hemi-face. Lateral presentation reduced the effect of a central bias, but did not eliminate the left hemi-face bias, resulting in more evident emotion specific viewing patterns and greater visual exploration of the face. Future work utilising eye tracking methodology with facial categorisation may consider carefully the impact of stimulus screen position and the effect of screen centre or left hemi-face biases.

### Funding

The authors received no funding for this work.

### Competing Interests

The authors declare there are no competing interests.

### Author Contributions

- Christopher J. Luke conceived and designed the experiments, performed the experiments, analyzed the data, contributed reagents/materials/analysis tools, wrote the paper, prepared figures and/or tables.
- Petra M.J. Pollux conceived and designed the experiments, analyzed the data, reviewed drafts of the paper.

### Human Ethics

The following information was supplied relating to ethical approvals (i.e., approving body and any reference numbers):

The experiment was granted ethical approval as part of a PhD project from the School of Psychology Research Ethics Committee (Soprec) at the University of Lincoln, conforming to British Psychological Society standards.

### Data Availability

Data and code openly accessible at Figshare:

Luke C. 2016. Lateral Presentation Alters Overall Viewing Strategy.

Figshare: https://dx.doi.org/10.6084/m9.figshare.3126766.v1.

### Supplemental Information

Supplemental information for this article can be found online at http://dx.doi.org/10.7717/peerj.2241#supplemental-information.

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
