# Peer review of "Lateral presentation of faces alters overall viewing strategy"

_PeerJ, doi:10.7717/peerj.2241_

## Round 0.1 · original submission · Major Revisions

Both reviewers highlight the potential interest of your manuscript in the perceptual laterality and the face perception communities, and I believe that they have good reasons to say so - the study presented is nice and clever and the implications are relevant to the field. Both reviewers, however, point to specific aspects in the description of the methodology and especially the statistical analyses, that need some important revision work of the present manuscript. I am sure you will be able to undertake the work on the suggested integrations and clarifications, improving the soundness and strength of your work.

Reviewer 1 ·

Basic reporting

Review of the manuscript entitled “Lateral presentation alters overall viewing strategy” by Christopher Luke and Petra Pollux:

The Authors investigated the possible interaction among types of gaze biases (a: center-of-face bias; b: left-gaze bias), emotional facial expressions (happy, sad and fear) presented at different intensities (from 0% to 100%, morphed with a neutral expression), and spatial position of the stimuli (center, left visual field: LVF, right visual field: RVF). Participants (21 females) had to recognize the emotional expression of faces, while eye movement were recorded. The results confirmed both biases during central presentation, and showed that the central bias was strongly weakened when stimuli were presented laterally, implying that the bias could be due to the “nested bias” to perceive the computer screen as an object, and thus to fixate its centre. Moreover, considering the ROIs constituted by eyes, nose, and mouth, the results confirmed the central bias only for nose, whereas eyes were fixated more frequently when faces were presented in the RVF than centrally, and mouth was fixated more frequently when presented either in the LVF or in the RVF than centrally. In particular, eyes were fixated more frequently in the RVF than in central presentation for sad and fear expressions. On the other hand, the results showed that the left-bias was stronger for stimuli presented in the RVF than for stimuli presented centrally, but it disappeared when they were presented in the LVF; conversely the percentage of a right hemi-face bias was lower for stimuli shown in the RVF than centrally, but it was higher for stimuli presented in the LVF with respect to the central presentation. In particular, the left-bias was lower for faces presented in the LVF than centrally for all of the 3 emotional expressions, and it was lower for faces presented centrally than in the RVF for happy and sad expressions. Emotion intensity influenced the results.

The manuscript is well-written and the theoretical focus of the study is very interesting. Nevertheless, I have a number of concerns, mainly concerning statistical analyses:

Experimental design

1) The first thing I noticed in the manuscript is that possible gender difference have not been considered. Why the Authors preferred to “avoid a possible gender bias”, and thus decided to test only female participants? If a gender bias exists, it would be interesting to discuss it. Similarly, 2 female and 2 male faces were used as stimuli, but the gender of the stimuli has not been considered as a possible factor in the ANOVAs. Could the Authors explain why? I believe that if the Authors really preferred to avoid a possible gender bias, maybe they would have selected only the photos of female actresses as experimental stimuli and only female participants (and/or male faces and male participants). I can suggest that the fact that females usually better recognize emotional expressions could be a more appropriate reason for their choice. See, for example:
Wang, B. (2013). Gender difference in recognition memory for neutral and emotional faces. Memory, 21(8), 991-1003.
Alternatively, could the Authors consider sex of the stimuli as an additional factor in their analysis?

2) In the same vein, the Authors stated that they created 11 levels of morphing between each emotional expression and the neutral expression (from 0% to 100%). Then, they specified that 7 different levels have been used (excluding 0%, 70%, 80%, and 90%). Why did they choose those 7 levels? The “non-symmetrical” choice can appear strange: one could expect that, for instance, 0%, 20%, 40%, 60% and 80% can be excluded. Why this unexpected (and apparently unjustified) choice?

Validity of the findings

3) One of the main weaknesses in the analysis is that the Left-bias has been considered by means of two separate analyses, carried out on the percentage of fixations to the left/right sides of the face. Why? I really believe that a whole ANOVA including “Hemi-face” (left, right) would have been more informative here. Could the Authors appropriately justify their choice? In alternative, could they repeat the analysis by including the factor Hemi-face? In my opinion, a direct statistical comparison between left-side and right-side would be very important for the purpose of the study (and it could simplify the results).

4) Similarly to the previous point, I also believe that a direct comparison among Emotions would be important. For example, concerning the Left-bias, the Screen position X Emotion interaction was significant. However, by means of pairwise comparisons, the Authors ignored the possible differences among emotions at each specific position and only reported the differences among positions for each emotional expressions. The same reasoning might be applied to the interaction among Screen position, Emotion and Intensity, and - in the Central-bias analysis – for Screen position X Emotion interaction.

5) I also believe that the results on the choice RTs should be analyzed. If – for instance –participants were more rapid at correctly recognizing emotional faces presented in the LVF than in the RVF or centrally, this could be another evidence of the left-bias. Anyway, all possible significant results concerning the RTs should be reported and discussed.

6) In different occurrences, the Authors stated that it has been previously shown that free viewing during the presentation of facial stimuli eliminates the left-bias. The Authors justified the difference between their results and those by Samson et al. (2014) referring to this aspect, concluding that the left-bias they found should be ascribed to the emotional categorization required in their task. Nevertheless, in this regard, I suggest to see (and cite), for instance:
Levy, J., Heller, W., Banich, M. T., & Burton, L. A. (1983). Asymmetry of perception in free viewing of chimeric faces. Brain and Cognition, 2(4), 404-419
Voyer, D., Voyer, S. D., & Tramonte, L. (2012). Free-viewing laterality tasks: A multilevel meta-analysis. Neuropsychology, 26(5), 551
I also suggest that the following manuscript, in which emotional faces were presented centrally and laterally and an emotional evaluation were required (supporting a right hemispheric superiority, and thus left-bias) could be cited:
Prete, G., D'Ascenzo, S., Laeng, B., Fabri, M., Foschi, N., & Tommasi, L. (2015). Conscious and unconscious processing of facial expressions: Evidence from two split‐brain patients. Journal of Neuropsychology, 9(1), 45-63

7) Were all participants right-handers? How was their handedness quantified? The Authors should also specify whether they asked participants to use a specific hand to carry out the task, and whether the hand used had been counterbalanced among participants.

Additional comments

8) The visual degrees of the ROIs of face viewing were defined for eyes (2 degrees) and nose (2 degrees), but this information is not reported for the mouth. Please, include this information.

9) In Figure 1, it would be better to invert the position of Central/Left presentation, in order to create a match between the presentation position and the columns position in the figure (and thus to facilitate the readers).

10) I suggest that Figure 2 should show the interaction between Screen position and Emotion, instead of the results in the 3 Screen positions, for left/right fixations (mainly if the Authors did not report the statistics between left and right, and so the readers cannot know if the difference between the gray and the black columns are significant or less).

Reviewer 2 ·

Basic reporting

I found this the writing contained in the paper to be of a very good standard, with a tight and well constructed introduction and discussion.

Experimental design

This is an intelligent design, containing research which will be of interest to a broad range of groups. The findings are well made and generally clearly expressed. I have some comments which I feel will improve the overall clarity of the manuscript which i will outline below.

Validity of the findings

No Comments

Additional comments

I enjoyed reading this well written paper. A few aspects are worthy of comment or action from the authors.

1. The procedure implies the response method was designed to obtain accurate RT’s, yet these are not provided. Could the authors provide some comment on why these were discarded from the report? Readers may find data on lateral judgement rts in particular of interest.
2. Regions of interest seem to be very different from 1-4, I feel some comment on this is warranted, 5 in particular seems to accommodate the whole face. Will this information be provided as a supplementary figure?
3. I am somewhat confused by the meaning of, P4 l 152: ‘fixations within the image but not within any ROI were included in percentage fixation calculations’ , could the authors please clarify, as no mention of other regions is subsequently made.
4. There are two occasions within the discussion where some clarity regarding the analysis the text is based upon should be given.

P6 l 241 It is stated ‘In contrast, the left-gaze bias for faces (Guo, 2012) was not solely attributable to general screen biases as left-gaze persisted regardless of stimulus screen position.’
I feel the authors should refer to the statistics that support this statement here. A reader may be confused by an examination of Figure 2 regarding left screen position, and the statement on line 269

P6 l 235 the authors state ‘Our results support the idea that the screen is treated as an object given that fixations were drawn to the centre of the screen regardless of the presented stimuli. ‘
Again here the authors should state what analysis this is referring to, is this in the context of centrally presented faces?



Other points
1. Stimuli size should be given in degrees, in line with other aspects of method
2. Model numbers selected as stimuli should be given
3. P5 l 198 suggest change left-face to left-hemi face for clarity, and same with right face, and at other points when referring to hemi-faces

---

## Round 0.2 · accepted · Accept

As you can see from the comments reported below, both reviewers are satisfied with your revision and recommend publication.

Reviewer 1 ·

Basic reporting

I believe that the changes and replies by the Authors fully address my
concerns, and I suggest that the paper is published in its present form.

Experimental design

ok

Validity of the findings

ok

Additional comments

no comments

Reviewer 2 ·

Basic reporting

No comments

Experimental design

No comments

Validity of the findings

No comments

Additional comments

I thank the authors for their timely attention to the points made by both my fellow reviewer and myself.